# Healthcare utilization among pre-frail and frail Puerto Ricans

**Cheyanne Barba**[1☉¤]*, **Brian Downer**[2☉], **Olivio J. Clay**[1‡], **Richard Kennedy**[3‡], **Erin Ballard**[1‡], **Michael Crowe**[1‡]

**1** Department of Psychology, University of Alabama at Birmingham, Birmingham, Alabama, Unites States of America, **2** Department of Nutrition, Metabolism, and Rehabilitation Sciences, University of Texas Medical Branch, Birmingham, Alabama, Unites States of America, **3** Division of Gerontology, Geriatrics, & Palliative Care, University of Alabama at Birmingham, Birmingham, Alabama, Unites States of America

☉ These authors contributed equally to this work.
¤ Current address: Mental Health Care Line, Michael E. DeBakey Veterans Affairs Medical Center, Houston, Texas, Unites States of America
‡ OJC, RK, EB and MC also contributed equally to this work.
* cbarba@uab.edu

**Data Availability Statement:** All PREHCO files are available from the Data Sharing for Demographic Research database (https://doi.org/10.3886/ICPSR34596.v1).

## Abstract

Frailty is associated with adverse health outcomes and greater healthcare utilization. Less is known about the relationship between frailty and healthcare utilization in Puerto Rico, where high rates of chronic conditions and limited healthcare may put this group at a higher likelihood of using healthcare resources. This study examined the association between pre-frailty and frailty with healthcare utilization at baseline and 4-year follow-up among a cohort of community dwelling Puerto Ricans living on the island. We examined data from 3,040 Puerto Ricans (mean age 70.6 years) from The Puerto Rican Elderly: Health Conditions (PREHCO) study between 2002–2003 and 2006–2007. We used a modified version of the Fried criteria defined as 3 or more of the following: shrinking, weakness, poor energy, slowness, and low physical activity. Pre-frailty was defined as 1–2 components. The number of emergency room visits, hospital stays, and doctor visits within the last year were self-reported. Zero-inflated negative binomial regression models were used for ER visits and hospital stays. Negative binomial models were used for doctor visits. Pre-frailty was associated with a higher rate of doctor visits with a rate ratio of 1.11 (95% CI = 1.01–1.22) at baseline. Frailty was associated with a higher rate of ER visits (1.48, 95% CI = 1.13–1.95), hospital stays (1.69, 95% CI = 1.08–2.65), and doctor visits (1.24, 95% CI = 1.10–1.39) at baseline. Pre-frailty and frailty were not associated with any healthcare outcomes at follow-up. Pre-frailty and frailty are associated with an increased rate of healthcare services cross-sectionally among Puerto Rican adults, which may cause additional burdens on the already pressured healthcare infrastructure on the island.

**Funding:** This work was supported by the National Institute on Aging (Awarded to MC: R01 AG064769 and Awarded to CB: R01AG064769-02S1 Supplement). The funders had no role in study design, data collection and analysis, decision to publish, or preparation of the manuscript.

**Competing interests:** The authors have declared that no competing interests exist.

## Introduction

As the population of Puerto Rico ages, understanding and treating health issues such as frailty become increasingly more important. Frailty is a syndrome that encompasses declines in multiple bodily systems and results in decreased physical functioning [1]. Research has suggested that Latinos have a greater prevalence of frailty [2] and Puerto Ricans in particular have high rates of disability [3], activity limitations [4], and chronic conditions [5] that may put them at a greater risk of becoming frail. Frailty is associated with adverse outcomes including falls [6], increased healthcare utilization [7], and mortality [8].

The population of adults aged 65 years and over in Latin America and the Caribbean is expected to rapidly increase to 15.5% by 2040 [9]. Puerto Rico specifically has experienced a rapid growth in the aging population on the island. Compared to the trajectories of six other countries with the highest population of 65+ years, Puerto Rico's older adult population grew the fastest after 2010 [10]. The age structure changes in Puerto Rico are mainly due to outward migration of working age individuals and low birth rates [10]. The growth in the older adult population has the potential to worsen the strain on the healthcare system on the island, especially given the greater morbidity of older Puerto Ricans compared to older adults on the mainland [11]. On the U.S. mainland, Puerto Ricans were 68% more likely than Mexican Americans and Central American/Caribbeans to have ambulatory care visits, 18% more likely to have an ER visit, and 30% more likely than non-Latino Whites to have an ER visit [12]. However, Puerto Ricans on the island may have different rates of healthcare utilization due to the rapidly aging population and unique characteristics of the healthcare system.

Previous work has found that diabetes is associated with frailty [13] and frail older adults were more likely to have diabetic complications [14]. Puerto Ricans have a higher prevalence of diabetes compared to older adults living on the mainland [15], poorer management of diabetes compared to Latinos in the U.S. [16], and diabetes was the second leading cause of death in Puerto Rico in 2019 [17]. Presence of diabetes among older Puerto Ricans may put the older adult population at a greater risk of complications related to frailty, leading to greater healthcare utilization. Regarding cognitive functioning, neurodegenerative diseases and memory impairment were found to be associated with increased healthcare utilization among older Mexican Americans [18] and a longitudinal cohort of older Americans [19], respectively. A recent study found that Puerto Ricans are more likely to report subjective cognitive impairment compared to non-Latino Whites [20]. As both frailty and lower cognitive functioning are both associated with greater healthcare utilization, we might expect frail Puerto Ricans with lower cognitive functioning to utilize emergency services at a higher rate than their robust counterparts.

Although Puerto Ricans receive Medicare after age 65, the quality of care on the island is severely lacking. A study of older Puerto Ricans enrolled in Medicare Advantage found inadequacies in 15 out of 17 components of diabetes care compared to Latino enrollees on the U.S. mainland [16]. Additionally, Medicaid funding and coverage of expenditures is significantly compared to residents on the mainland [21]. Health centers on the island also face technical and financial barriers to providing care including debt [22], crowded emergency rooms [22], and an outward flow of medical professionals [23].

Among Latinos generally, use of healthcare resources varies across subgroups due to insured status, primary language spoken, country of origin, and length of time lived in the U.S. [12] Studies among Mexican Americans found that frailty was associated with increased Medicare spending [24] and Mexican Americans who were pre-frail or frail had significantly higher risk for hospitalizations compared to those who were non-frail [25]. While Latinos share many

cultural experiences, there are differences among Latino subgroups in important areas such as ethnic/racial backgrounds, acculturation, colonial influences, socioeconomic status, and health status. The unique experience of Puerto Ricans on the island should be considered in the context of debilitating health conditions and the effect on healthcare utilization. Puerto Rico has suffered economic hardships due to the history of economic influence and colonial rule by the mainland U.S. and the healthcare system is one in particular that has faced many struggles [26, 27]. There is a lack of funding for healthcare and limitations in coverage for Medicare and Medicaid [21, 27] in Puerto Rico compared to the mainland U.S. Characterizing frailty and the potential impact on healthcare use in Puerto Rico may help clarify the potential increases in healthcare services, distribution of resources across the island, and the potential financial and practical support needed from families.

An additional unique characteristic of the island is the rapidly aging population. A greater number of older adults increases the presence of debilitating conditions such as frailty which in turn could increase healthcare utilization, causing even greater strains on the healthcare system. Studying the prevalence of frailty and utilization of resources among frail adults can help inform allocation of funding for Medicare and Medicaid as well as targeted interventions on potential comorbidities of frailty. At the time of this study, there does not appear to be any studies examining the association of frailty and healthcare utilization among older Puerto Ricans living on the island. The aim of this study is to fill the gap in the literature on the utilization of healthcare among pre-frail and frail older Puerto Ricans.

## Materials and methods

### Participants

Participants were from the Puerto Rican Elderly: Health Conditions (PREHCO) study, which is a longitudinal survey of 4,291 noninstitutionalized Puerto Rican adults aged 60 years and older in Puerto Rico [28]. Wave 1 data collection began in 2002–2003 and Wave 2 was completed in 2006–2007. Participants were recruited door-to-door based on census data and maps of the municipalities. Interviews at both time points were conducted in Spanish and included information regarding participants' self-reported medical history, demographics, early life factors, psychosocial information, and a cognitive screener. The PREHCO study included households with at least one adult aged 60 years or older and included households with multiple family units. For participants with baseline cognitive impairment (based on minimental Cabán score <11), a proxy informant completed an abbreviated questionnaire regarding late life health and functional status. The study excluded adults that were institutionalized and adults living on the islands of Vieques and Culebra. The selection of the analytic sample is shown in Fig 1. From the initial target sample (n = 4,291), participants who could not complete the initial interview were excluded (n = 578) and a total of 3,713 participants completed the baseline interview. Those that were missing data on any of the variables of interest (n = 301), including those that needed a proxy, or missing greater than 1 frailty item (n = 372) at baseline were excluded from all analyses. This left a total of 3,040 of non-frail (n = 479), pre-frail (n = 1,754), and frail (n = 807) participants to evaluate descriptive characteristics and analyses of healthcare utilization. By 4-year follow-up a total of 306 participants died, 110 were lost to follow up, 93 refused to participate, 24 were institutionalized, and 12 did not participate for other reasons. For ER visits and hospital stays, 55 were missing information and 101 were missing doctor visit information. The final analytic sample was 2,440 for ER and hospital stay analyses and 2,394 for doctor visit analyses.

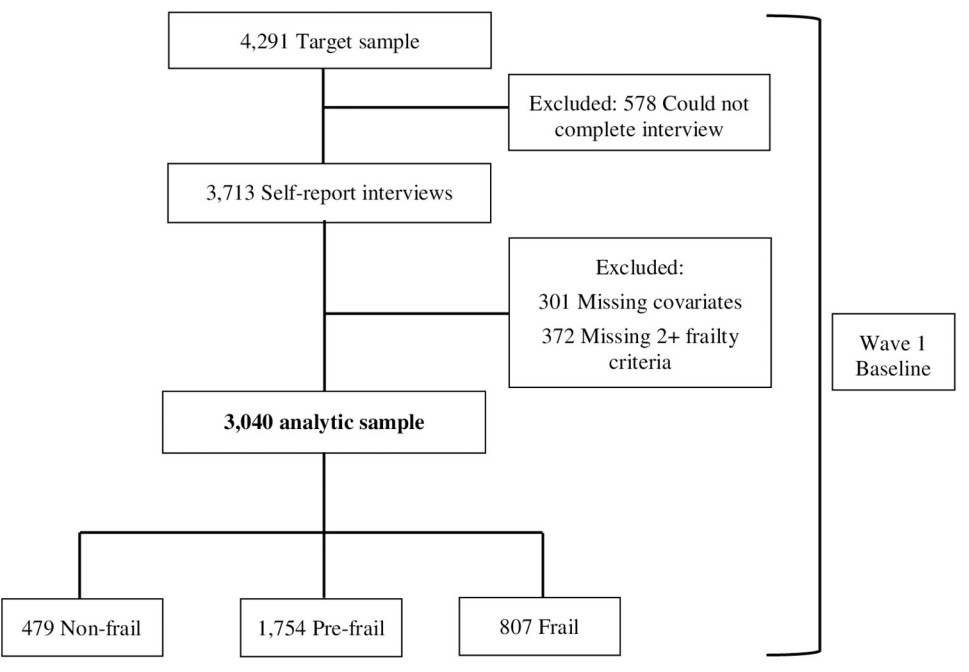

**Fig 1. Selection of the analytic sample.**

## Measures

**Outcome: Healthcare utilization.** Healthcare utilization was measured by self-reported number of emergency room visits, nights hospitalized, and total number of doctor office visits within the last 12 months.

**Frailty.** Frailty was measured by using a modified version of the Fried et al. criteria of shrinking, weakness, poor energy, slowness, and low physical activity [29]. We defined frailty at baseline as a summed variable including: 1) *Shrinking*: self-reported unintentional weight loss of greater than 10lbs over the last year; 2) *Weakness*: inability to stand on one-leg for 10 seconds [30, 31]; 3) *Poor energy*: measured by the Geriatric Depression Scale (GDS) question, "Do you feel full of energy?" and coded as positive when respondents responded 'no'; 4) *Slowness*: measured by the Timed Up and Go (TUG) which has participants stand up from a chair without using their arms, walk 3 meters, turn and walk back, and sit down again [32]. The slowest 20% of the group, adjusted for sex and height, was considered slow [29]; 5) *Low physical activity*: answering 'no' to the question, "In the last year, have you engaged in any of the following activities: sports, jogging, walking, dancing, or heavy labor, three or more times a week?" Frailty criteria are outlined in Table 1 and criteria for frailty at follow-up is available in S1 Table.

Grip strength was not available at baseline and leg stand was used as a proxy measure. The one-leg stand has been shown to predict fall risk among frail individuals [33] and those categorized as frail performed worse on the one-leg stand [34]. Additionally, the one-leg stand and grip strength at follow-up were highly positively correlated ($r = .91, p < .0001$). Previous work showing an association between the one-leg stand with frailty and the high correlation between the one-leg stand and grip strength in this study provided support that the one-leg stand could serve as a proxy measure. Interviewees unable to perform the one-leg stand or the timed up due to safety or physical concerns were coded as positive for weakness [35]. This study did not have a measure of only walking speed as used in Fried et al. [29]. We utilized the TUG, which

**Table 1. Frailty criteria characteristics for the PREHCO analytic sample (n = 3,040) at baseline (2002–2003).**

| Criteria | N (%) | Missing frailty items |
|---|---|---|
| **Shrinking (weight loss)** | | |
| **Yes** | 661 (21.7) | |
| **No** | 2,379 (78.3) | |
| **Weakness (leg stand)** | | |
| **Yes** | 1,765 (58.1) | |
| **No** | 1,275 (41.9) | |
| **Full of energy** | | |
| **Yes** | 2,499 (82.2) | |
| **No** | 541 (17.8) | |
| **Slowness (timed up and go)** | | 108 |
| **<20th percentile** | 738 (25.2) | |
| **>20th percentile** | 2,194 (74.8) | |
| **Physical activity** | | |
| **Yes** | 1,352 (44.5) | |
| **No** | 1,688 (55.5) | |

The summed frailty variable included: 1) *Shrinking*: self-reported unintentional weight loss of greater than 10lbs over the last year, 2) *Weakness*: inability to stand on one-leg for 10 seconds, 3) *Poor energy*: measured by the GDS question, "Do you feel full of energy?" and coded as positive when respondents responded 'no', 4) *Slowness*: the slowest 20% of the group, adjusted for sex and height, was considered slow measured by the TUG, 5) *Low physical activity*: answering 'no' to the question, "In the last year, have you engaged in any of the following activities: sports, jogging, walking, dancing, or heavy labor, three or more times a week?"

is a measure of gait function, same as walking speed. Both walking speed and TUG were found to be highly correlated in a population with a high rate of physical disability [36]. If participants were missing one item of frailty criteria a score was assigned based on the sum of the reported four criteria [37]. Criteria were summed and then categorized into one of three stages: (0) non-frail, (1–2) pre-frail, (3–5) frail.

**Cognitive functioning.** Cognitive functioning was measured using the minimental Cabán (MMC). The MMC was created to be more appropriate for use in Latino populations compared to the Spanish version of the Mini-Mental State Exam (MMSE) [38]. The MMC included measures of orientation, immediate and delayed verbal recall, visual memory, copy of intersecting pentagons, clock drawing, and comprehension of a three-step command. Total scores range from 0–20 points with higher scores indicating better cognitive performance.

**Diabetes.** Diabetes was self-reported, and participants were asked "has a doctor ever told you that you have diabetes, that is, high levels of sugar in your blood" (yes/no).

**Covariates.** Participants self-reported age (years), sex (male/female), race, and level of education (number of school years completed). Options for racial identification included: Black, Multiracial/Trigueño/a (mixed black and white/wheat colored), White, Mestizo/a (European/White and Indigenous), and other. A summed index score of self-reported vascular diseases was created including hypertension, myocardial infarction, congestive heart failure, and stroke/transient ischemic attack (TIA). BMI was calculated by weight and height taken by the interviewer. BMI $\geq 30 \, \text{kg/m}^2$ was considered obese (yes/no). Depressive symptoms were assessed using a modified Spanish language version 15-item Geriatric Depression Scale (GDS) [39]. Participants were asked yes or no questions regarding their mood within the past 2 weeks. The GDS is on a 0–15 scale and higher scores reflect greater depressive symptoms.

The Katz Index of activities of daily living (ADL) included difficulty walking, transferring, eating, bathing, toileting, or dressing due to a health problem [40]. The Lawton Index of instrumental activities of daily living (IADL) included difficulties with using the telephone, transportation, shopping, preparing meals, household chores, managing finances, and medications [41]. The number of ADL/IADL limitations were summed together, and an impairment score (range 0–13) was created with higher scores equivalent to greater impairment [42].

Health insurance was categorized into one of four groups: Medicaid (reference group), Medicare (part A only, parts A&B), non-Medicare/private (Triple S, Blue Cross, Humana, pension plans), or uninsured/missing. To measure barriers to care, participants were asked "in the last two years, have you needed medical attention that you could not get" and "why were you unable to get the medical attention that you needed" (response options: lack of transportation, it was too time consuming, could not pay for them, did not consider it a serious problem, health plan would not cover them, appointment was scheduled for later, "other," did not have doctor's referral, had no authorization from the health plan). Household size was measured as living alone or with others (yes/no).

## Statistical analysis

Descriptive statistics compared demographic characteristics across frailty states using analysis of variance (ANOVA) for continuous variables and $\chi^2$ for categorical variables. T-tests and $\chi^2$ analyses were used to compare participants included in the analysis to excluded participants missing greater than 1 frailty item. To determine model fit, we first checked the dispersion of the data to determine the use of Poisson regression or a negative binomial model. The Pearson Chi-Square equaled 2.7, 6.1, and 7.1 for longitudinal emergency room visits, hospital stays, and doctor's visits, respectively. Results suggested that Poisson was not an adequate model fit because the data were overdispersed. Upon further examination of the data with frequency plots, figures revealed over 50% of zero counts for emergency room visits and over 70% of zero counts for hospital stays. Then we considered the underlying process to explain why participants had zero count data. Participants with frailty symptoms had health insurance and the potential to utilize healthcare services, but other barriers may exist to account for the zero counts such as reduced access to healthcare facilities due to location or long wait times that resulted in cancelling of appointments. Therefore, zero counts are considered a result of more than one process [43].

ER visits and length of hospital stay data were analyzed with a zero-inflated negative binomial model and compared to a zero-inflated Poisson model. The Akaike Information Criterion (AIC) can be compared across both models to determine the best fit [43]. The AIC for the zero-inflated negative binomial model was lower by more than 2 (a difference of at least 2 indicates better fit) compared to the zero-inflated Poisson model, further emphasizing a better model fit with the negative binomial model (Fig 2). Regarding doctor visits, the original Poisson model showed a high Pearson Chi-Square indicating a poor model fit. Frequency plots of doctor visits did not support an excess of zero counts (<10% zero counts). Therefore, a negative binomial regression fit the data better [43].

Additionally, frequency plots were examined for predictors and covariates to determine which variables might predict the probability of a zero count. The following were added to ZERMODEL statements for (1) hospital stays at baseline: vascular disease score, depressive symptoms, and ADL/IADL disability, (2) ER visits at follow-up: vascular disease score, and (3) hospital stays at follow-up: vascular disease score.

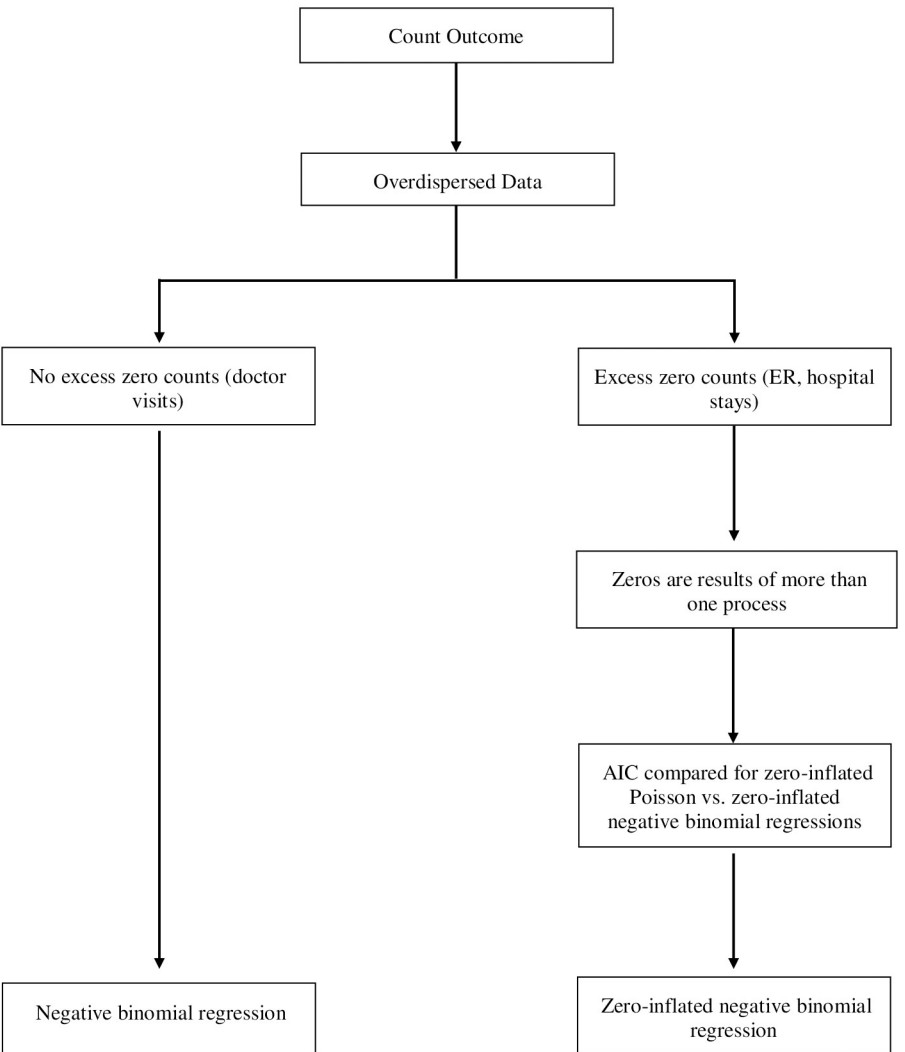

**Fig 2. Statistical decision tree.** Decision process for determining which GLM method is appropriate for overdispersed data.

Regression models were initially unadjusted. Then, cross-sectional and longitudinal models were adjusted for age, sex, race, and education. In the final models, ADL/IADL score, cognitive functioning, obesity, depressive symptoms, diabetes, the summed vascular disease score, living situation, insurance coverage, and barriers to care were added. Logistic regression models examined the association of frailty with death by follow-up. In order to identify whether associations were moderated by diabetes and cognitive functioning, regression analyses were performed with a model including interaction terms between frailty status and diabetes and a separate model examining frailty status and cognitive functioning. Models with interaction terms included all covariates. Subjects' participation was voluntary and informed consent was obtained both verbally and in written form. If a participant was unable to complete the interview due to cognitive impairment or any other reason a spouse, adult child, or relative was interviewed as a proxy. The IRB at the University of Alabama at Birmingham approved the current study (IRB# 300004744). All data analyses were performed in SAS, version 9.4 [44].

## Results

### Sample characteristics

At baseline, participant mean age was 70.6 years and 58.6% of participants were female. The average education level was 8.5 years, mean MMC score was 16.7 points, and 26.2% of participants reported a diagnosis of diabetes (Table 2). A total of 15.8% of the sample was non-frail,

**Table 2. Descriptive information for the PREHCO cohort at baseline 2002–2003 (n = 3,040).**

| Variable | Total (n = 3,040) | Non-frail (n = 479) | Pre-frail (n = 1,754) | Frail (n = 807) | p-value |
|---|---|---|---|---|---|
| **ER visits, N (%)** | 944 (31.1) | 107 (22.3) | 489 (27.9) | 348 (43.1) | < .0001 |
| **Hospital stays, mean (SD)** | 1.6 (6.2) | 0.7 (4.8) | 1.1 (4.7) | 3.2 (8.9) | < .0001 |
| **Doctor visits, N (%)** | 2,646 (87.0) | 388 (81.0) | 1,514 (86.3) | 744 (92.2) | < .0001 |
| **Age, mean (SD)** | 70.6 (7.9) | 67.1 (5.7) | 70.1 (7.6) | 73.6 (8.5) | < .0001 |
| **Sex, N (%)** | | | | | < .0001 |
| Female | 1,780 (58.6) | 186 (38.8) | 1,057 (60.3) | 537 (66.5) | |
| Male | 1,260 (41.5) | 293 (61.2) | 697 (39.7) | 270 (33.5) | |
| **Education, mean (SD)** | 8.5 (4.8) | 10.4 (4.4) | 8.7 (4.7) | 7.0 (4.7) | < .0001 |
| **Race, N (%)** | | | | | .19 |
| Black | 172 (5.7) | 39 (8.1) | 101 (5.8) | 32 (4.0) | |
| Trigueño/a | 1,238 (40.7) | 188 (39.3) | 711 (40.5) | 339 (42.0) | |
| Mestizo/a | 267 (8.8) | 42 (8.8) | 784 (44.7) | 67 (8.3) | |
| White (ref) | 1,363 (44.8) | 210 (43.8) | 784 (44.7) | 369 (45.7) | |
| **ADL/IADL, mean (SD)** | 0.7 (1.6) | 0.1 (0.5) | 0.5 (1.2) | 1.6 (2.3) | < .0001 |
| **Obese, N (%)** | | | | | .03 |
| Yes | 850 (28.0) | 110 (23.0) | 508 (29.0) | 232 (28.8) | |
| No | 2,190 (72.0) | 369 (77.0) | 1,246 (71.0) | 575 (71.3) | |
| **Depressive symptoms, mean (SD)** | 3.1 (3.3) | 1.5 (1.8) | 2.6 (2.8) | 5.2 (3.9) | < .0001 |
| **Cognitive scores, mean (SD)** | 16.7 (2.4) | 17.2 (2.2) | 16.8 (2.3) | 16.2 (2.5) | < .0001 |
| **Diabetes, N (%)** | | | | | < .0001 |
| Yes | 797 (26.2) | 91 (19.0) | 432 (24.1) | 283 (35.1) | |
| No | 2,243 (73.8) | 388 (81.0) | 1,331 (75.8) | 524 (64.9) | |
| **Vascular risk factors, N (%)** | | | | | < .0001 |
| 0 | 1,143 (37.6) | 225 (47.0) | 704 (40.1) | 214 (26.5) | |
| 1 | 1,318 (43.4) | 201 (42.0) | 789 (45.0) | 328 (40.6) | |
| 2 | 407 (13.4) | 38 (8.0) | 192 (11.0) | 177 (21.9) | |
| 3 | 151 (5.0) | 12 (2.5) | 66 (3.8) | 73 (9.1) | |
| 4 | 21 (0.7) | 3 (0.6) | 3 (0.2) | 15 (1.9) | |
| **Type of insurance, N (%)** | | | | | < .0001 |
| Medicaid (ref) | 1,312 (43.2) | 162 (33.8) | 749 (42.7) | 401 (49.7) | |
| Medicare | 1,026 (33.8) | 147 (30.7) | 584 (33.3) | 295 (36.6) | |
| Non-Medicare/Private | 609 (20.0) | 150 (31.3) | 365 (20.8) | 94 (11.7) | |
| Uninsured/Missing | 93 (3.1) | 20 (4.2) | 56 (3.2) | 17 (2.1) | |
| **Barriers, N (%)** | | | | | .01 |
| Yes | 132 (4.4) | 14 (92.9) | 68 (3.9) | 50 (6.2) | |
| No | 2,904 (95.7) | 465 (97.1) | 1,685 (96.1) | 754 (93.8) | |
| **Living alone, N (%)** | | | | | < .0001 |
| Yes | 961 (31.6) | 113 (23.6) | 542 (30.9) | 306 (37.9) | |
| No | 2,079 (68.4) | 366 (76.4) | 1,212 (69.1) | 501 (62.1) | |

ADL/IADL functioning represents difficulties in activities of daily living and instrumental activities of daily living. ANOVA was used for continuous variables and Chi-square for categorical variables.

**Table 3. Unadjusted and adjusted rate ratios (95% CI) for ER visits, hospital stays, and doctor's visits at baseline (2002–2003) by frailty status (n = 3,040).**

| Variable | Pre-frail | | | Frail | | |
|---|---|---|---|---|---|---|
| | Model 1 | Model 2 | Model 3 | Model 1 | Model 2 | Model 3 |
| **ER visits** | 1.43 (1.13–1.81) | 1.39 (1.09–1.77) | 1.25 (0.99–1.58) | 2.06 (1.60–2.65) | 2.08 (1.60–2.70) | 1.48 (1.13–1.95) |
| **Hospital stays** | 1.12 (0.75–1.66) | 1.33 (0.88–2.00) | 1.27 (0.84–1.93) | 1.70 (1.14–2.53) | 2.06 (1.35–3.15) | 1.69 (1.08–2.65) |
| **Doctor visits** | 1.24 (1.12–1.37) | 1.23 (1.11–1.35) | 1.11 (1.01–1.22) | 1.71 (1.53–1.90) | 1.69 (1.50–1.89) | 1.24 (1.10–1.39) |

ER = emergency room; Model 1 is unadjusted. Model 2 included age, sex, race, and education. Model 3 added ADL/IADL score, baseline cognition, obesity, depressive symptoms, diabetes, the summed vascular disease score, insurance coverage, barriers to care, and living situation.

57.7% pre-frail, and 26.5% frail. Most participants had Medicaid (43.2%) or Medicare (33.8%) as their primary insurance. The most common reasons for not receiving medical attention were not being able to pay for the services (23.1%), not having a doctor's referral (20%), health insurance would not cover it (14.6%), or another reason (26.2%). Frail participants had on average a greater number of ER visits, length of hospital stays, and doctor visits, compared to pre-frail and non-frail participants. At follow-up, within the last year of the interview date there were a total of 895 ER visits, average length of hospital stays was 0.4 nights (SD = 1.7), and there were 2,185 doctor's visits. Cross-sectionally, those excluded for missing more than 1 frailty item (n = 372) were older, more likely to be female and less educated, and reported more ER visits, longer hospital stays, and more doctor's visits (all $p < .05$).

**Association of frailty with healthcare utilization at baseline.** Pre-frailty was associated with an increased rate of ER visits compared to non-frail participants at a rate ratio of 1.43 (95% CI = 1.13–1.81) in the unadjusted Model 1 (Table 3). After inclusion of initial demographic covariates, the rate ratio decreased and the association was statistically significant (1.39, 95% CI = 1.09–1.77). The inclusion of all covariates in Model 3 accounted for the previous association of pre-frailty with ER visits and the model was no longer statistically significant (1.25, 95% CI = 0.99–1.58). Pre-frailty was not associated with hospital stays in any models ($p > .05$). There was a significant association between pre-frailty and doctor visits in the fully adjusted model, with an increased rate of 1.11 (95% CI = 1.01–1.22). In models that included the interaction term of frailty status and diabetes, there was a significant interaction effect on rate of ER visits (0.55, 95% CI = 0.33–0.91). Pre-frail participants with diabetes had a decreased rate of ER visits. There were no significant interactions between pre-frailty and hospital stays or doctor visits. Interactions between pre-frailty and cognitive functioning were not statistically significant.

Frailty was associated with an increased rate of ER visits compared to non-frail participants in the fully adjusted Model 3, with a rate ratio of 1.48 (95% CI = 1.13–1.95). Frailty was associated with length of hospital stays after adjusting for all covariates, (1.69, 95% CI = 1.08–2.65). Frailty was associated with an increased rate of doctor visits with a rate ratio of 1.24 (95% CI = 1.10–1.39). In models examining the interaction of frailty status with diabetes, frail participants with diabetes were at a decreased rate of hospital stays (0.37, 95% CI = 0.16–0.88). Interactions examining frailty status by cognitive functioning were not statistically significant.

## Baseline frailty status and healthcare use at 4-year follow-up

The association between pre-frailty at baseline and ER visits at follow-up was not statistically significant in the fully adjusted model (Table 4). Pre-frailty was not related to length of hospital stays in Model 3 (0.97, 95% CI = 0.68–1.37). In the initial model, pre-frailty was associated with an increased rate of doctor visits (1.11, 95% CI = 1.01–1.23), but this relationship was attenuated after adjustment for covariates in subsequent models.

**Table 4. Unadjusted and adjusted rate ratios (95% CI) for ER visits, hospital stays, and doctor's visits at follow-up (2006–2007) by baseline frailty status.**

| Variable | Pre-frail | | | Frail | | |
|---|---|---|---|---|---|---|
| | Model 1 | Model 2 | Model 3 | Model 1 | Model 2 | Model 3 |
| ER visits | 1.18 | 1.08 | 0.95 | 1.78 | 1.53 | 0.95 |
| (n = 2,440) | (.096–1.45) | (0.88–1.33) | (0.77–1.17) | (1.42–2.23) | (1.20–1.95) | (0.74–1.23) |
| Hospital stays | 1.07 | 1.11 | 0.97 | 2.04 | 2.04 | 1.42 |
| (n = 2,440) | (0.77–1.45) | (0.79–1.57) | (0.68–1.37) | (1.43–2.90) | (1.39–2.98) | (0.94–2.15) |
| Doctor visits | 1.11 | 1.07 | 1.01 | 1.31 | 1.25 | 0.99 |
| (n = 2,394) | (1.01–1.23) | (0.97–1.19) | (0.91–1.12) | (1.17–1.47) | (1.10–1.41) | (0.87–1.13) |

ER = emergency room; Model 1 is unadjusted. Model 2 included age, sex, race, and education. Model 3 added ADL/IADL score, baseline cognition, obesity, depressive symptoms, diabetes, the summed vascular disease score, insurance coverage, barriers to care, and living situation.

Baseline frailty was initially associated with ER visits in the unadjusted Model 1 (1.78, 95% CI = 1.42–2.23). However, the association was no longer statistically significant after adjusting for health covariates. Frailty was associated with an increased rate of hospital stays in Model 1 (2.04, 95% CI = 1.43–2.90). After adjustment for covariates in subsequent models, the association was no longer statistically significant. Frailty was not related to doctor visits in the fully adjusted Model 3 (0.99, 95% CI = 0.87–1.13).

## Discussion

This study examined the association between frailty and healthcare utilization among pre-frail and frail Puerto Ricans. Frailty is an important issue to address among Puerto Ricans as the number of older adults on the island is growing rapidly and the healthcare system is already struggling to support the population amidst low funding, lack of physicians, and setbacks caused by hurricanes and the pandemic. In this study we found that pre-frailty was associated with an increased rate of doctor visits and frailty was associated with an increased rate of ER visits, hospital stays, and doctor visits cross-sectionally. This is consistent with other studies that have found increased healthcare utilization among frail adults that may be attributable to declines in physical health, higher disease burden, and social circumstances [45–48]. However, we found no association between frailty and healthcare utilization across the 4-year follow-up. This may be because frailty has a greater association with mortality than healthcare utilization over time. Statistical models examining the association of frailty at baseline with death by follow-up showed that frailty was significantly associated with a 1.35 times increased rate of mortality over the 4-year follow-up period in this sample (2.35, 95% CI = 1.46–3.78). An additional factor to consider is that the loss of frail subjects may have reduced the overall rates of healthcare resources by follow-up given that the relatively healthiest of the cohort survived over the 4 years. Another possibility is that families are caring for frail older adults when injuries do occur. In a study of frailty across 8 countries, the percentage of frail older adults in Puerto Rico was comparable to other Latin American countries but the proportion of frail older adults that were dependent on family as caregivers was greatest in Puerto Rico [49]. This may mean that if frail older adults are ill, families may be providing care as opposed to immediately visiting the ER, especially if these services are costly or inaccessible.

At baseline, 15.8% of our sample was non-frail, 57.7% were pre-frail, and 26.5% were classified as frail. This is consistent with other studies reporting on frailty among Latino populations. A meta-analysis of community-dwelling adults over the age of 60 years reported a prevalence of 19.6% with a range of 7.7% to 42.6% in 29 Latin American and Caribbean

countries [2]. The prevalence of frailty in the literature is variable, most likely due to various definitions of frailty.

Although previous research has found Puerto Ricans to have poorly managed diabetes [13, 50], we found that pre-frail diabetic participants had a decreased rate of ER visits and frail diabetic participants had a decreased rate of hospital stays. This is inconsistent with previous research that has found frail participants with diabetes have greater utilization of healthcare resources [7, 51]. This may be a unique feature to island-dwelling Puerto Ricans because of the limited healthcare services provided by Medicaid and Medicare and the limited healthcare infrastructure on the island. The decreased utilization of health resources in this sample may reflect the greater issue of poorly managed diabetes on the island. Medicare beneficiaries on the island receive worse healthcare and receive less disease-modifying treatment [16] compared to Hispanic and White Medicare enrollees in the U.S.

Contrary to our hypothesis, lower cognitive functioning did not moderate the relationship between frailty and healthcare utilization. In a study of older Mexican American Medicare beneficiaries, participants with Alzheimer's disease or a related dementia (ADRD) had a higher odds of hospitalizations and ER visits than those without ADRD [18]. Other studies have found that frailty alone, and not the combination of cognitive impairment and frailty, is a better predictor of death [52]. It may be that for this cohort, frailty above cognitive functioning is a better predictor of healthcare utilization. Additionally, the design of the PREHCO study excluded participants with cognitive impairment at baseline to observe incident cognitive changes over time. This may have excluded participants that were both cognitively impaired and frail, reducing the likelihood of an association with healthcare utilization.

The average hospital stay among frail participants was 3.2 nights. The average length of stay among our sample is lower than reported by other studies. A meta-analysis of non-frail, pre-frail, and frail Brazilian residents aged 60 years and older reported an average length of stay of 5.0–17.9 nights among frail older adults [53]. A possible explanation for why this sample might have shorter hospital stays is the profit structure of hospitals on the island. More than half of hospitals in Puerto Rico are for-profit while a little over a quarter of hospitals on the mainland are for-profit [27]. Profit structure and low coverage of hospital services from insurance companies might influence a patient's length of stay. Additionally, there is limited hospital care in rural areas, a smaller percentage of large hospitals, and a lower rate of beds per resident compared to the mainland U.S. which may increase the discharge rate [27]. Despite the relatively lower hospital length stay among our sample, frailty significantly increased the rate of hospital stays compared to non-frail participants, emphasizing the need to screen and care for fail participants in order to decrease the need for expensive medical services.

## Strengths, limitations, and future research

The strengths of this study include the use of longitudinal data, recruitment of a community sample, and examination of a subgroup of Latinos. A potential limitation of this study is the self-report of healthcare utilization. Although one study found that self-report of ER visits, hospital admissions, and doctor visits within a year generally correspond with objective utilization data [54]. Another limitation of this study is the inability to exactly replicate Fried's frailty criteria. This study used the one-leg stand as a proxy measure for weakness. Although different, the one-leg stand has been shown to predict fall risk [33] and those who are frail perform worse on the one-leg stand [34]. Additionally, since research suggests Puerto Ricans have greater disability and activity limitations, our more mobility focused definition of frailty may be a more culturally adapt definition for this population [55]. Future research should examine multiple definitions of frailty to distinguish possible cultural/ethnic differences among diverse

groups. The current study did not examine additional barriers to care such as geographical distance to healthcare facilities. Future studies could examine the differences in utilization of healthcare between rural and urban residents on the island with the consideration of socioeconomic differences and limitations in access to healthcare.

## Conclusion

This is the first study to our knowledge to examine the association of frailty with healthcare use among older island-dwelling Puerto Ricans. We contribute to a growing literature on characterizing subgroups of Latinos that are frequently overlooked or grouped together despite differences in health characteristics and healthcare utilization. Our findings suggest that pre-frailty and frailty are associated with increased rates of ER visits, hospital stays, and doctor visits within the last year but were not associated with increased healthcare use longitudinally. Our findings suggest there may be other population specific factors to consider such as the role of caregivers and access to healthcare services. Additional research is warranted to characterize the health status of Puerto Ricans and to predict health outcomes in this group with the goal of intervening on potentially modifiable risk factors.

## Supporting information

**S1 Table. Frailty criteria characteristics at baseline (2002–2003) and follow-up (2006–2007).**
(DOCX)

## Acknowledgments

We would like to thank A. Garcia and A-L. Dávila for data collection and management.

## Author Contributions

**Conceptualization:** Cheyanne Barba, Brian Downer, Olivio J. Clay, Richard Kennedy, Michael Crowe.

**Data curation:** Cheyanne Barba.

**Formal analysis:** Cheyanne Barba.

**Funding acquisition:** Michael Crowe.

**Methodology:** Cheyanne Barba, Brian Downer, Olivio J. Clay, Michael Crowe.

**Project administration:** Michael Crowe.

**Resources:** Brian Downer, Richard Kennedy.

**Writing – original draft:** Cheyanne Barba.

**Writing – review & editing:** Cheyanne Barba, Brian Downer, Olivio J. Clay, Richard Kennedy, Erin Ballard, Michael Crowe.

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
