## [Decision Letter · Decision Letter 0]

25 Aug 2022

PONE-D-22-18720Healthcare utilization among pre-frail and frail Puerto RicansPLOS ONE

Dear Dr. Barba,

Thank you for submitting your manuscript to PLOS ONE. After careful consideration, we feel that it has merit but does not fully meet PLOS ONE’s publication criteria as it currently stands. Therefore, we invite you to submit a revised version of the manuscript that addresses the points raised during the review process.

This topic is interesting to the Gerontology field, although the manuscript has several weaknesses in all the sections of this. I suggest attending the comments of both reviewers, mainly the observations of the reviewer 1 to improve the manuscript. It is necessary to attach a point-to-point letter if you want to resend the manuscript.

We look forward to receiving your revised manuscript.

Kind regards,

Martha Asuncion Sánchez-Rodríguez, PhD

Academic Editor

PLOS ONE

Journal Requirements:

"This work was supported by the National Institute on Aging (R01 AG064769 and R01AG064769-02S1)".

 "This work was supported by the National Institute on Aging (Awarded to MC: R01 AG064769 and Awarded to CB: R01AG064769-02S1 Supplement). The funders had no role in study design, data collection and analysis, decision to publish, or preparation of the manuscript".

Additional Editor Comments:

This topic is interesting to the Gerontology field, although the manuscript has several weaknesses in all the sections of this. I suggest attending the comments of both reviewers, mainly the observations of the reviewer 1 to improve the manuscript. It is necessary to attach a point-to-point letter if you want to resend the manuscript.

Reviewers' comments:

Reviewer's Responses to Questions

**Comments to the Author**

1. Is the manuscript technically sound, and do the data support the conclusions?

Reviewer #1: No

Reviewer #2: Yes

2. Has the statistical analysis been performed appropriately and rigorously? 

Reviewer #1: No

Reviewer #2: Yes

3. Have the authors made all data underlying the findings in their manuscript fully available?

Reviewer #1: No

Reviewer #2: Yes

4. Is the manuscript presented in an intelligible fashion and written in standard English?

Reviewer #1: Yes

Reviewer #2: Yes

5. Review Comments to the Author

Reviewer #1: The objective of the study was to determine the association between pre-frailty and frailty with healthcare utilization at baseline and 4-year follow-up among a cohort of community dwelling Puerto Ricans living on the island. This topic is of great relevance in the fields of study of Gerontology and Geriatrics. However, the manuscript has too many weaknesses in scientific support, method, results, discussion and conclusions.

Major comments

1. Introduction

The authors must include the scientific support that justifies their study. In this sense, they have to highlight the novelty of their research:

• Are there not studies on this issue in Puerto Ricans or in other countries?, that allow specifying the relevance of their study.

• What is the relevance of knowing the ealthcare utilization among pre-frail and frail Puerto Ricans.

• What implications (economic, quality or adequacy of the type of care) does it have for state and private health services, society, the family and the patient, considering the possible greater use of health services among pre-frail and frail Puerto Ricans?

2. Method

• The authors should specify how they determined the sample size, especially because of the sample size of the subgroups (non-frail n=479; pre-frail n= 1754 and frail n= 807).

• Was the sample size the same in the second wave (2006-2007) as in the first wave (2002-2003)?, was the number and proportion of the groups the same in both waves?

•The greatest limitation of the study is the measurement criteria used for the diagnosis of pre-frail and frail, for example: “Poor energy: measured by the Geriatric Depression Scale (GDS) question, “Do you feel full of energy?”

3. Results

• The authors must include a table regarding the criteria related to the frailty that the subjects had by group at the beginning of the second measurement.

• It is also convenient to include a table regarding the health care services used by the subjects by group, in order to carry out an analysis of its implications in the discussion.

4. Discussion

• The discussion must be rewritten, since the results are not analyzed with respect to their impact and social and economic implications.

• The analysis is contradictory, the authors point out that there is no increase in the use of healthcare services and refer that it coincides with other studies that did find it. “However, we found no association between frailty and healthcare utilization longitudinally. This is consistent with other studies that have found increased healthcare utilization among frail adults which can be attributed to declines in physical health, higher disease burden, and social circumstances" (lines 305 to 308).

4. Concusion

• The conclusions are not supported by the findings.

The general study does not provide new knowledge, especially because the focus and analysis is limited. In this sense, if the results are negative (the use of healthcare services does not increase), these should be analyzed in depth, highlighting that perhaps the reason is because the family is responsible for said healthcare.

Reviewer #2: I. Major claims of the manuscript: 1) In community dwelling Porto Ricans living in the Island, prefrailty and frailty were associated with an increased rate of health care utilization, cross-sectionally. 2) Frailty was not significantly related to long term health care utilization in this study.

II. Literature cited: The authors quoted relevant previous literature on the association of frailty and health care utilization in differente mainland and Puerto Rico Latino populations.

III. The combination of crss-sectional and longitudinal research designs, as well as the statistical treatment of the data fully support the claims

IV. However, contrary to the original hypothesis of the association of frailty with increased health care utilization, the research results show the opposite. The explanation offered by the authors relies heavily on the low quality and cost of the Puerto Rico Health System. It is my impression that there are other contributing factors that could help to understand the sociocultural context of the study. For instance, what is the proportion of rural subjects in the baseline sample? This would be a point to discuss since several barriers to health care are to be found in rural Municipalities where economic conditions are probably lower than in the urban ones. For instance, geographical access and distance. Besides, the health infrastructure in the rural areas has been heavily damaged by recent hurricans. Another point to consider is the higher mortality of subjects during the follow-up: has this loss of frail subjects of the cohort had an impact on the results? One more point: differential access to the health services by sex. There were more women than men in the sample. Has this difference had an effect on the results?

These points, among others, would help to understand why frailty was not associated with higher rate of health care utilization in the follow-up, and to guide new research to explain this finding.

6. PLOS authors have the option to publish the peer review history of their article (what does this mean?). If published, this will include your full peer review and any attached files.

Reviewer #1: **Yes: **Víctor Manuel Mendoza-Núñez

Reviewer #2: No

---

## [Author Response · Author response to Decision Letter 0]

23 Oct 2022

Reviewer #1: The objective of the study was to determine the association between pre-frailty and frailty with healthcare utilization at baseline and 4-year follow-up among a cohort of community dwelling Puerto Ricans living on the island. This topic is of great relevance in the fields of study of Gerontology and Geriatrics. However, the manuscript has too many weaknesses in scientific support, method, results, discussion and conclusions.

Major comments

1. Introduction

The authors must include the scientific support that justifies their study. In this sense, they have to highlight the novelty of their research:

• Are there not studies on this issue in Puerto Ricans or in other countries?, that allow specifying the relevance of their study.

We have clarified that there are no studies, that the authors are aware of, that examine the association of frailty with healthcare utilization in Puerto Rico. The purpose of this study is to fill this gap among Puerto Ricans. Research with Mexican Americans has found that frailty was associated with increased hospitalizations and Medicare spending. However, other studies have shown there are differences in healthcare utilization among Latino subgroups. This provides support for our study to examine the rates of healthcare utilization among Puerto Ricans who live on the island (vs. the mainland U.S.) which may present another difference in the rates of utilization. This information has been added to the introduction. 

• What is the relevance of knowing the ealthcare utilization among pre-frail and frail Puerto Ricans.

We have clarified information regarding the relevance of knowing healthcare utilization among pre-frail and frail Puerto Ricans in the introduction. This issue is relevant because the older adult population of Puerto Rico is aging rapidly and therefore the increase in frailty may also rise. This is an issue because the healthcare infrastructure on the island is lacking in funding and in resources. The increases in frailty among older citizens has the potential to cause increases in healthcare utilization, causing further strain on the healthcare system. Characterizing the prevalence of frailty and utilization of resources among frail adults can help inform allocation of funding for Medicare and Medicaid (which is significantly lower in proportion despite the need of residents who are older and of lower socioeconomic status) as well as targeted interventions on potential comorbidities of frailty. 

• What implications (economic, quality or adequacy of the type of care) does it have for state and private health services, society, the family and the patient, considering the possible greater use of health services among pre-frail and frail Puerto Ricans?

Information on the state of the quality of care in Puerto Rico is provided in the introduction, highlighting the inadequacies in care, lack of funding, financial barriers, and the exodus of medical professionals from the island. We also provide information regarding the poor-quality care for diabetes, a potential moderator of healthcare use, and cognitive functioning, which is relevant for this population because of their rapid aging, vascular risk factors, and greater reports of subjective cognitive impairment. 

2. Method

• The authors should specify how they determined the sample size, especially because of the sample size of the subgroups (non-frail n=479; pre-frail n= 1754 and frail n= 807).

Sample size is clarified in the participants section, and we have added additional details of the outcome sample. 

From the initial target sample (n=4,291), participants who could not complete the initial interview were excluded (n=578) and a total of 3,713 participants completed the baseline interview. Those that were missing data on any of the variables of interest (n=301), including those that needed a proxy, or missing greater than 1 frailty item (n=372) at baseline were excluded from all analyses. This left a total of 3,040 of non-frail (n=479), pre-frail (n=1,754), and frail (n=807) participants to evaluate descriptive characteristics and analyses of healthcare utilization. By 4-year follow-up a total of 306 participants died, 110 were lost to follow up, 93 refused to participate, 24 were institutionalized, and 12 did not participate for other reasons. For ER visits and hospital stays, 55 were missing information and 101 were missing doctor visit information. The final analytic sample was 2,440 for ER and hospital stay analyses and 2,394 for doctor visit analyses.

The subgroups of frailty are determined by the frailty criteria outlined under the Frailty heading. An additional table (now Table 1) was added to detail the frailty criteria used and display the number of subjects in each criterion. Additionally, a Supplemental Table 1 has been added that includes both baseline frailty criteria and follow-up frailty criteria and corresponding numbers of participants in each criterion. 

• Was the sample size the same in the second wave (2006-2007) as in the first wave (2002-2003)?, was the number and proportion of the groups the same in both waves?

Clarification was added to the Methods regarding Wave 2 sample size and additional boxes were added to the figure to outline missing participants and the final analytic sample. The outline of sample size is included in the above comment for your convenience. 

•The greatest limitation of the study is the measurement criteria used for the diagnosis of pre-frail and frail, for example: “Poor energy: measured by the Geriatric Depression Scale (GDS) question, “Do you feel full of energy?”

We do agree that not using the exact Fried et al. criteria is not ideal. However, there is not an established definition of frailty in the literature and the Fried criteria is held as a gold standard but not an official definition. Given this, there is some flexibility in the literature with using proxy criteria. We also believe that the benefits of including this population in the literature, addressing the issues facing the aging population on the island, and encouraging other researchers to consider diverse populations in the conversation around aging outweigh the limitation of the current definition. There are also previous studies that have included the GDS (instead of the CES-D) in their definition of frailty. The Fried et al. criteria originally proposed use of two self-report questions to measure exhaustion from the CES-D which state “I felt that everything I did was an effort” and “I could not get going.” Other studies examining frailty have used the GDS before as well (Ensrud et al., 2020) while other studies have used other depression symptom inventories (Rodriguez et al., 2018). As there is no set definition of frailty in the literature and both the CES-D and the GDS are quite similar symptom inventories, we feel the exhaustion question in the GDS is a comparable and valid proxy to the CES-D criteria that Fried et al. originally proposed. 

3. Results

• The authors must include a table regarding the criteria related to the frailty that the subjects had by group at the beginning of the second measurement.

A table (Table 1) was added to the Methods section outlining the baseline frailty criteria with the number of subjects with each criterion. As the aim of this manuscript is not examining frailty status at follow-up, we included the criteria for frailty at follow-up as a supplementary table and corrected language in the methods under the frailty descriptor to describe frailty at baseline only. 

• It is also convenient to include a table regarding the health care services used by the subjects by group, in order to carry out an analysis of its implications in the discussion.

Healthcare services used by each frailty group was included in the descriptive table (now Table 2) but was relocated to the top of the table for convenience. 

4. Discussion

• The discussion must be rewritten, since the results are not analyzed with respect to their impact and social and economic implications.

More information was added to the discussion to elaborate on the social and economic associations with our findings. Namely, we did not find an association between frailty and healthcare utilization longitudinally. We elaborate that frailty may be a better predictor of mortality which is one possibility. It may also be that families are caring for older adults when they become ill. The costliness of ER services and the inaccessibility of healthcare are noted as possible contributors to reduced healthcare use. We also describe the economic structure of the hospital system on the island and how this system does not benefit the patient. We also discuss limited hospital care in rural areas, the smaller number of large hospitals across the island, and lower number of beds available per resident on the island which may influence the decreased hospital stays in this sample. 

• The analysis is contradictory, the authors point out that there is no increase in the use of healthcare services and refer that it coincides with other studies that did find it. “However, we found no association between frailty and healthcare utilization longitudinally. This is consistent with other studies that have found increased healthcare utilization among frail adults which can be attributed to declines in physical health, higher disease burden, and social circumstances" (lines 305 to 308).

We have clarified our wording to properly state that pre-frailty was associated with outcomes cross-sectionally at baseline and that this is consistent with the literature. However, the association of frailty over the 4 years with increased healthcare utilization was not significant. We further elaborate on reasons why this association was not significant (see explanation above which is also relevant to this comment). 

4. Concusion

• The conclusions are not supported by the findings. The general study does not provide new knowledge, especially because the focus and analysis is limited. In this sense, if the results are negative (the use of healthcare services does not increase), these should be analyzed in depth, highlighting that perhaps the reason is because the family is responsible for said healthcare.

We appreciate this feedback. We have considered your comments and have rewritten the conclusion to better reflect our contributions to the literature and reflect on why we did not find associations longitudinally. 

Reviewer #2: I. Major claims of the manuscript: 1) In community dwelling Porto Ricans living in the Island, prefrailty and frailty were associated with an increased rate of health care utilization, cross-sectionally. 2) Frailty was not significantly related to long term health care utilization in this study.

II. Literature cited: The authors quoted relevant previous literature on the association of frailty and health care utilization in differente mainland and Puerto Rico Latino populations.

III. The combination of crss-sectional and longitudinal research designs, as well as the statistical treatment of the data fully support the claims

IV. However, contrary to the original hypothesis of the association of frailty with increased health care utilization, the research results show the opposite. 

• The explanation offered by the authors relies heavily on the low quality and cost of the Puerto Rico Health System. It is my impression that there are other contributing factors that could help to understand the sociocultural context of the study. For instance, what is the proportion of rural subjects in the baseline sample? This would be a point to discuss since several barriers to health care are to be found in rural Municipalities where economic conditions are probably lower than in the urban ones. For instance, geographical access and distance. Besides, the health infrastructure in the rural areas has been heavily damaged by recent hurricans. 

Differences in rural and urban residents on the island are important as the reviewer suggests. However, approximately 6% of the population in Puerto Rico is considered rural during the time of data collection and presently (https://data.worldbank.org/indicator/SP.RUR.TOTL.ZS?locations=PR&most_recent_value_desc=false) . The rural population has steadily declined since the 1960’s as various agricultural crops have declined over time. We have added this as a limitation to this study and also offer this concept as a suggestion for future research.

• Another point to consider is the higher mortality of subjects during the follow-up: has this loss of frail subjects of the cohort had an impact on the results? 

We have added analysis and discussion on the impact of mortality in the discussion. Frailty was associated with a 2 times increased rate of mortality over the follow-up period and the loss of subjects may have reduced the overall rate of healthcare utilization since we only measured healthcare rates at follow up (not every year over the course of the 4 years). Therefore, we may only be examining the relatively healthier individuals in this cohort who survived and may be using less healthcare resources, leading to our null results. The frail subjects who died may have been more likely to use healthcare resources. 

• One more point: differential access to the health services by sex. There were more women than men in the sample. Has this difference had an effect on the results?

This is a great point. Sex differences do exist in healthcare, usually dependent on socioeconomic status or differences in use of healthcare. We examined interactions of frailty status and sex with each of the outcomes of this study and did not find any statistically significant differences (p>.05) between the groups. 

These points, among others, would help to understand why frailty was not associated with higher rate of health care utilization in the follow-up, and to guide new research to explain this finding.

References cited:

Ensrud, K. E., Kats, A. M., Schousboe, J. T., Taylor, B. C., Vo, T. N., Cawthon, P. M., Hoffman, A. R., Langsetmo, L., & Study, O. F. i. M. (2020). Frailty Phenotype and Healthcare Costs and Utilization in Older Men. J Am Geriatr Soc, 68(9), 2034-2042. https://doi.org/https://doi.org/10.1111/jgs.15381

Rodriguez, J. J. L., Prina, A. M., Acosta, D., Guerra, M., Huang, Y., Jacob, K., Jimenez-Velasquez, I. Z., Salas, A., Sosa, A. L., & Williams, J. D. (2018). The prevalence and correlates of frailty in urban and rural populations in Latin America, China, and India: a 10/66 population-based survey. Journal of the American Medical Directors Association, 19(4), 287-295. e284. https://doi.org/https://doi.org/10.1016/j.jamda.2017.09.026

---

## [Decision Letter · Decision Letter 1]

23 Nov 2022

PONE-D-22-18720R1Healthcare utilization among pre-frail and frail Puerto RicansPLOS ONE

Dear Dr. Barba,

Thank you for submitting your manuscript to PLOS ONE. After careful consideration, we feel that it has merit but does not fully meet PLOS ONE’s publication criteria as it currently stands. Therefore, we invite you to submit a revised version of the manuscript that addresses the points raised during the review process.

The manuscript improved substantially. The topic is of interest for the gerontologic community. I suggest accepting the comments of reviewer 1.

We look forward to receiving your revised manuscript.

Kind regards,

Martha Asuncion Sánchez-Rodríguez, PhD

Academic Editor

PLOS ONE

Journal Requirements:

Additional Editor Comments (if provided):

The manuscript improved substantially. The topic is of interest for the gerontologic community. I suggest accepting the comments of reviewer 1.

Reviewers' comments:

Reviewer's Responses to Questions

**Comments to the Author**

1. If the authors have adequately addressed your comments raised in a previous round of review and you feel that this manuscript is now acceptable for publication, you may indicate that here to bypass the “Comments to the Author” section, enter your conflict of interest statement in the “Confidential to Editor” section, and submit your "Accept" recommendation.

Reviewer #1: All comments have been addressed

Reviewer #2: All comments have been addressed

2. Is the manuscript technically sound, and do the data support the conclusions?

Reviewer #1: Yes

Reviewer #2: Yes

3. Has the statistical analysis been performed appropriately and rigorously? 

Reviewer #1: Yes

Reviewer #2: Yes

4. Have the authors made all data underlying the findings in their manuscript fully available?

Reviewer #1: Yes

Reviewer #2: Yes

5. Is the manuscript presented in an intelligible fashion and written in standard English?

Reviewer #1: Yes

Reviewer #2: Yes

6. Review Comments to the Author

Reviewer #1: The authors responded to comments and significantly improved the manuscript.

Minor comments

1. Please, I suggest that you remove the last paragraph of the introduction, since it is irrelevant and the hypothesis is obvious. “This study will examine the association of frailty and utilization of ER visits, hospital stays, and doctor visits in a population of older Puerto Ricans living on the island. We expect that pre-frailty and frailty will be associated with greater utilization of healthcare services and this association will be moderated by diabetes and lower cognitive functioning”.

2. Please, it is necessary to correct, in the Discussion section, the paragraph “Statistical models examining the association of frailty at baseline with death by follow-up showed that frailty was significantly associated with a 2.35 times increased rate of mortality over the 4-year follow-up period in this sample (2.35, 95% CI = 1.46-3.78)”. It should say “1.35 times increased rate”. In this sense, remember that the value of RR=1 is equal, and therefore, the value of "1" must be subtracted for the interpretation.

Reviewer #2: This research uncovers singularities and limitations of the Puerto Rican Island Health System, especially on the care of an aging population. It would be of interest whether further research clarifies the reasons behind the lack of association of health care with frailty.

7. PLOS authors have the option to publish the peer review history of their article (what does this mean?). If published, this will include your full peer review and any attached files.

Reviewer #1: **Yes: **Víctor Manuel Mendoza-Núñez

Reviewer #2: No

---

## [Author Response · Author response to Decision Letter 1]

4 Dec 2022

References were reviewed and edited. Two references were deleted due to restructuring of the discussion. These were reviewed and re-cited in the text where relevant. See “Frailty is a syndrome that encompasses declines in multiple bodily systems and results in decreased physical functioning (1)” in the introduction and “Other studies have found that frailty alone, and not the combination of cognitive impairment and frailty, is a better predictor of death (52)” in the discussion. 

Reviewer #1: The authors responded to comments and significantly improved the manuscript.

Minor comments

1. Please, I suggest that you remove the last paragraph of the introduction, since it is irrelevant and the hypothesis is obvious. “This study will examine the association of frailty and utilization of ER visits, hospital stays, and doctor visits in a population of older Puerto Ricans living on the island. We expect that pre-frailty and frailty will be associated with greater utilization of healthcare services and this association will be moderated by diabetes and lower cognitive functioning”.

The last paragraph in the introduction, as outlined above, has been removed. 

2. Please, it is necessary to correct, in the Discussion section, the paragraph “Statistical models examining the association of frailty at baseline with death by follow-up showed that frailty was significantly associated with a 2.35 times increased rate of mortality over the 4-year follow-up period in this sample (2.35, 95% CI = 1.46-3.78)”. It should say “1.35 times increased rate”. In this sense, remember that the value of RR=1 is equal, and therefore, the value of "1" must be subtracted for the interpretation.

Thank you for pointing this out. We have changed the above mentioned sentence. 

Reviewer #2: This research uncovers singularities and limitations of the Puerto Rican Island Health System, especially on the care of an aging population. It would be of interest whether further research clarifies the reasons behind the lack of association of health care with frailty.

The authors agree with this statement and believe the suggestion for future research to investigate this lack of association is addressed in the conclusion “Our findings suggest there may be other population specific factors to consider such as the role of caregivers and access to healthcare services. Additional research is warranted to characterize the health status of Puerto Ricans and to predict health outcomes in this group with the goal of intervening on potentially modifiable risk factors.”

---

## [Editor Report · Decision Letter 2]

21 Dec 2022

Healthcare utilization among pre-frail and frail Puerto Ricans

PONE-D-22-18720R2

Dear Dr. Barba,

We’re pleased to inform you that your manuscript has been judged scientifically suitable for publication and will be formally accepted for publication once it meets all outstanding technical requirements.

Kind regards,

Martha Asuncion Sánchez-Rodríguez, PhD

Academic Editor

PLOS ONE
---

## [Editor Report · Acceptance letter]

27 Dec 2022

PONE-D-22-18720R2 

Healthcare utilization among pre-frail and frail Puerto Ricans 

Dear Dr. Barba:

I'm pleased to inform you that your manuscript has been deemed suitable for publication in PLOS ONE. Congratulations! Your manuscript is now with our production department. 

Kind regards, 

on behalf of

Dr. Martha Asuncion Sánchez-Rodríguez 

Academic Editor

PLOS ONE